# Effect of Rejuvenating Agent on the Pavement Properties of Cold Recycled Mixture with Bitmen Emulsion

**Chun Li** [1], **Jian Ouyang** [2,*], **Peng Cao** [3], **Jingtao Shi** [1], **Wenting Yang** [4] **and Yuqi Sha** [5]

1   PetroChina Fuel Oil Limited Company Research Institute, Beijing 100195, China; li-chun@petrochina.com.cn (C.L.); sjt0822@163.com (J.S.)
2   School of Transportation and Logistics, Dalian University of Technology, Dalian 116024, China
3   College of Architecture and Civil Engineering, Beijing University of Technology, Beijing 100124, China; caopeng518888@126.com
4   School of Civil Engineering, Dalian University of Technology, Dalian 116024, China; 1773736603wt@mail.dlut.edu.cn
5   School of Energy and Powder Engineering, Xi'an Jiaotong University, Xi'an 710049, China; sha15399456173@126.com
*   Correspondence: ouyangjian@dlut.edu.cn or ouyangjian87@126.com

**Abstract:** A traditional cold recycled mixture with bitumen emulsion (CRMB) has a relatively low cracking resistance and moisture susceptibility, which greatly limits its application scope. A rejuvenating agent was employed to improve the pavement properties of CRMB. To avoid the rejuvenating agent having an adverse effect on the new bitumen, reclaimed asphalt pavement (RAP) was firstly treated by the rejuvenating agent, and the effect of rejuvenating time on the pavement properties of CRMB was investigated. Results indicate that the addition of a rejuvenating agent can greatly improve the ductility and moisture susceptibility of CRMB. Meanwhile, although the rejuvenating agent can soften aged bitumen, the addition of a rejuvenating agent can still increase the indirect tensile strength of CRMB and does not greatly reduce the rutting resistance of CRMB. This phenomenon exists because the rejuvenating agent can be both beneficial to the membrane structure of the bitumen emulsion mastic and aged bitumen. It can also greatly improve the bonding interface between RAP and the bitumen emulsion mastic. The rejuvenating time between RAP and the rejuvenating agent can affect the pavement properties of CRMB. To obtain better pavement properties, the optimum recommended rejuvenating time is between 5 and 7 days. Overall, the addition of rejuvenating agent can be a good choice to improve the pavement properties of CRMB.

**Keywords:** cold recycled mixture; bitumen emulsion; rejuvenating agent; pavement properties; rejuvenating time



## 1. Introduction

Environmental sustainability and cost-effectiveness are promoting the worldwide diffusion of low-energy and low-emission technologies for pavement construction and rehabilitation. In this context, the recycling of reclaimed asphalt pavement (RAP) based on a cold recycled mixture with bitumen emulsion (CRMB) is one of the most effective and low environmental impact technologies. CRMB consists of bitumen emulsion, additional water, filler, unheated virgin aggregates, and RAP. Compared to conventional hot-recycled bitumen mixture, the maximum amount of RAP, generally between 70% and 100% of the total aggregate blend, can be recycled in CRMB [1]. The effective utilization of RAP can greatly preserve natural resources and reduce the disposal costs. No extra heating is required in the production of CRMB, so it can minimize energy consumption, dust, and gaseous emissions [2]. As a result of these advantages, CRMB is becoming more preferred in pavement construction and rehabilitation.

Although CRMB has been widely applied in the pavement construction and rehabilitation, its application is mainly restricted to the base and subbase layers of motorways [3].

Therefore, the surplus value of RAP cannot be maximized by this technology. To better reuse RAP, road engineers in China have been trying to use CRMB as a surface layer material (mainly intermediate and bottom courses of asphalt layer) in recent years. However, there are many challenges to achieve this ambitious goal. The primary challenge is the mechanical properties of CRMB. CRMB has a high modulus and good rutting resistance [4–6] because cement hydrates can greatly increase the stiffness of bitumen binder [7]. However, because of the high void content and the addition of cement, CRMB features low tensile strength and low ductility [8,9], so that its cracking resistance is weak compared to conventional hot mix asphalt and hot recycling asphalt [10]. CRMB also has a high moisture susceptibility [11,12]. As a result, CRMB has high risks of cracking and moisture damage when it is employed in the surface layer. To reduce these risks, the pavement properties of CRMB should be greatly improved, especially for tensile strength, ductility, and moisture susceptibility.

To reduce the risks of cracking and moisture damage, an indirect tensile test is suggested to be better than the Marshall test in the mix design of CRMB [3,9]. Meanwhile, factors influencing the indirect tensile strength (ITS) and pavement properties of CRMB were intensively studied by scholars recently, such as cement content and cement types [13–15], other cementitious fillers and content [11,16], water content [9], bitumen content [15], aggregate gradation [17,18], curing condition [19–21], and compaction method [22]. These studies are all very meaningful to understand the effect of raw components and fabrication conditions on the ITS and pavement performances of CRMB and obtain the optimal mix design of CRMB. Comparing the results, the most efficient way to improve the tensile strength and moisture damage of CRMB is adding more cement into mixture. However, the deformability of CRMB can be greatly degraded if more cement is added [9], thus increasing the risk of cracking. The cement content of CRMB should be between 1% and 2% to prevent cracking in the application. Overall, the pavement properties of CRMB under the optimal mix proportion should be further improved to promote the use of RAP.

To further improve the pavement properties of CRMB under the optimal mix proportion, the deficiencies of CRMB in the mechanical properties should be firstly analyzed. There are significant differences between CRMB and hot recycling asphalt in the materials composition and RAP treatment method. Firstly, the hot bitumen with a very low viscosity can easily wet and coat aggregate well. However, bitumen droplets are difficult to penetrate to the surface texture of aggregates. Its coating ability on aggregate is highly related to the wetting ability of bitumen emulsion. In this context, the addition of surfactants (i.e., superplasticizer and wetting agent), which can greatly reduce the viscosity of fresh cement bitumen emulsion mastic and the contact angle between asphalt emulsion and aggregate, can greatly increase the ITS of CRMB and have no adverse effects on its ductility [8,23]. Secondly, although the RAP content is small in the hot recycling asphalt, a rejuvenating agent is employed to restore the technical properties of the aged bitumen of RAP in the technology [24,25]. However, no treatment method is employed for RAP, and no extra heating is used to melt the aged bitumen in the application of CRMB; thus, the RAP is normally as black aggregates in the traditional CRMB. The high amount of RAP is only adhered by a small amount of cement bitumen emulsion mastic. As a result of the poor ductility and adhesive ability of aged bitumen, CRMB has relative low pavement properties. To further improve the pavement properties of CRMB, adding a rejuvenating agent to restore the technical properties of aged bitumen may be a good choice according to the experience of hot recycling asphalt [26].

With the above regards, a rejuvenating agent is tried to improve the mechanical properties of CRMB in this study. Since the reactivity between the rejuvenating agent and aged bitumen at ambient temperature is much gentler than that at high temperature, more concerns should be focused on the treatment process of RAP by rejuvenating agent in CRMB. To efficiently restore the aged bitumen in RAP and avoid softening the new bitumen in cement bitumen emulsion mastic, a better treatment method may be to firstly

blend RAP and the rejuvenating agent and then cure this blending for several days. In consideration of this, the objectives of this paper are as follows:

- To study the effect of a rejuvenating agent on the pavement properties of CRBM;
- To find a suitable treatment process of RAP by rejuvenating agent;
- To discuss the possible mechanisms of the effect of the rejuvenating agent on the pavement properties of CRBM.

## 2. Materials and Specimens Preparation

### 2.1. Materials

Slow setting cationic bitumen emulsion, Portland ordinary cement P.O42.5, recycled asphalt pavement (RAP), crushed sand, and limestone powder were employed to fabricate CRMB. The Portland ordinary cement P.O42.5 is from Dalian Xiaoyetian Cement Co., Ltd. in Dalian, China. The properties of cationic bitumen emulsion are listed in Table 1. The employed RAP was obtained from the surface layer of one motorway in Liaoning Province, China, whose gradation is shown in Figure 1. The bitumen content in RAP was determined as 4.8% by combustion method. According to the gradation of RAP, virgin crushed sand, limestone powder, and cement were introduced to get a suitable aggregate gradation of CRMB, which is shown in Figure 1. RAP accounts for 80% of solids by weight in CRMB. Other solids (i.e., crushed sand, limestone powder, and cement) account for the remaining 20%. Cement and limestone powder contents were 1.5% and 1.5% for all mixtures, respectively.

**Table 1.** Properties of bitumen emulsion.

| Property | Value |
|---|---|
| Test on emulsion | |
| Emulsifier content (%) | 3.5 |
| Mean particle diameter (μm) | 1.520 |
| Sieve test (1.18 mm, %) | 0 |
| Storage stability (1 day, 25 °C,%) | 0.02 |
| Storage stability (5 days, 25 °C,%) | 0.6 |
| Mixing stability with cement, residual content above 1.18 mm (%) | 0.75 |
| Test on residue from distillation | |
| Solid content (%) | 57.0 |
| Penetration (25 °C) | 78.2 |
| Softening point (Ring and Ball method, °C) | 48.8 |
| Ductility (25 °C, cm) | 85 |

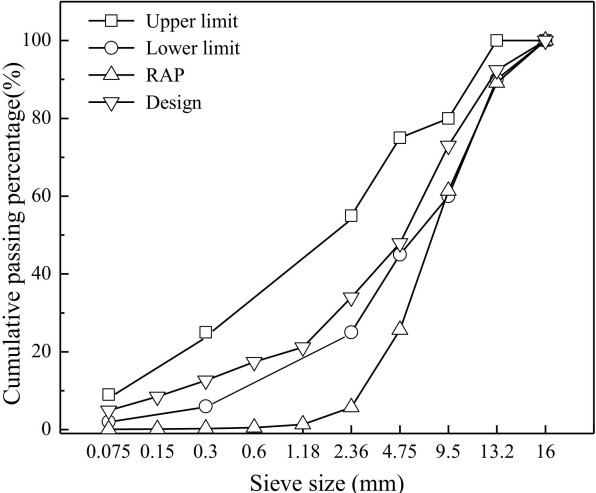

**Figure 1.** Gradation of CRMB.

A rejuvenating agent was employed to improve the pavement properties of CRMB. The main component of the employed rejuvenating agent is extract oil, which contains a large number of saturates and aromatics and can restore the technical properties of aged bitumen. Unlike the rejuvenating agent in hot recycling asphalt, the rejuvenating agent should be blended with RAP at ambient temperature in CRMB; thus, its viscosity at ambient temperature should be very low. Therefore, the employed rejuvenating agent was a high-concentrated emulsion material with a main component of extract oil.

### 2.2. Mix Design and Specimen Preparation

The function of the rejuvenating agent is to restore the technical properties of aged bitumen. Therefore, the rejuvenating agent should be firstly blended with RAP. However, the reactivity and molecular diffusion between the rejuvenating agent and aged bitumen at ambient temperature is much gentler than those at high temperature. If bitumen emulsion is added immediately after the rejuvenating agent blends well with RAP, the new bitumen in emulsion can be also softened by the rejuvenating agent. To reduce this adverse effect, the RAP should be firstly treated by the rejuvenating agent. Thus, the RAP was firstly blended well with the rejuvenating agent and then kept at 25 °C for several days. The reactivity time between RAP and the rejuvenating agent was chosen as 2, 5, 7, and 10 days in this study.

The optimum rejuvenating agent content, the total water, and new bitumen content of CRMB should be determined before studying the effect of rejuvenating agent on the pavement properties of CRMB. The three parameters of mix proportion were all determined by the maximum indirect tensile strength (ITS). Firstly, the effect of rejuvenating agent content on the ITS and void content of CRMB was investigated in Figure 2. CRMB has the maximum ITS when the rejuvenating agent content is 1%. Therefore, the optimum content of rejuvenating agent was determined as 1%. It can be seen from Figure 2 that the addition of rejuvenating agent can improve the compactability of CRMB. Secondly, the optimum total water content of CRMB with and without the rejuvenating agent was determined by the same way, which was 4.3% and 5.3%, respectively. Finally, under the optimum content of rejuvenating agent and total water, the optimum bitumen emulsion content of CRMB with and without the rejuvenating agent was determined as 3% and 4%, respectively. Therefore, the addition of rejuvenating agent can both reduce the water and bitumen content of CRMB.

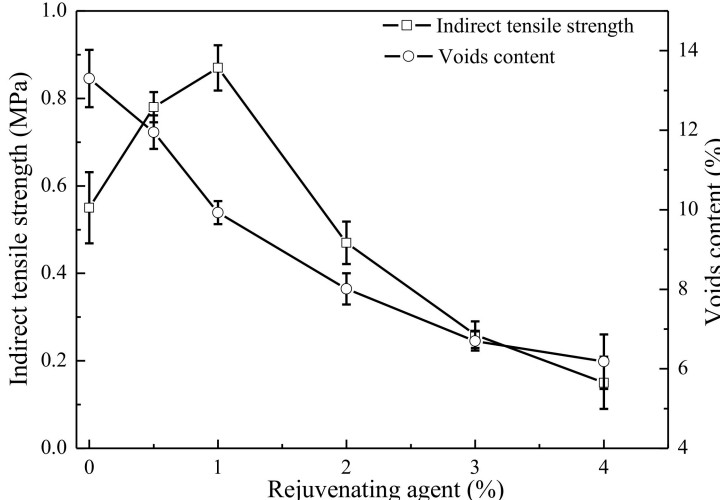

**Figure 2.** Effect of rejuvenating agent content on the ITS and void content of CRMB with bitumen emulsion content at 3% and additional water content at 3%.

The tested specimens were compacted by a Marshall hammer (from Shanghai Changji Geological Instrument Co., Ltd., Shanghai, China) with 75 blows per side in molds with a

diameter of 101.6 mm. After 1 day of curing in the mold, the specimens were demolded into an oven at a temperature of 60 °C for 2 days curing before testing.

## 3. Testing Programs

In this study, the volumetric and pavement properties of CRMB were firstly investigated, and then, some tests were performed to discuss the possible mechanisms of the effect of rejuvenating agent on the pavement properties of CRMB. Therefore, two parts of tests were performed in this study, including the basic pavement properties tests and the corresponding mechanisms analysis tests. For each test, at least three specimens were produced to determine the reproducibility of the results.

### 3.1. Pavement Properties Tests

3.1.1. Void Content Test

The void content of CRME is calculated according to Equation (1) [27].

$$VV = (1 - \frac{\rho_s}{\rho_t}) \times 100\% \tag{1}$$

where $\rho_s$ is the apparent specific gravity of the mixture; and $\rho_t$ is the maximum specific gravity of the mixture. The apparent specific gravity of CRMB can be measured by the surface-dry condition method after curing. The maximum specific gravity of CRMB can be calculated by Equation (2).

$$\rho_t = \frac{100 + P_a}{\frac{P_1}{\gamma_1} + \frac{P_2}{\gamma_2} + \frac{P_3}{\gamma_3} + \frac{P_4}{\gamma_4} + \frac{P_a}{\gamma_a}} \times \rho_w \tag{2}$$

where $P_a$ is the new bitumen aggregate ratio; $P_1$, $P_2$, $P_3$, and $P_4$ are the blending percentage of RAP, virgin aggregate, cement, and limestone powder, respectively; $\gamma_1$, $\gamma_2$, $\gamma_3$, and $\gamma_4$ are the corresponding specific gravity of aggregate; $\gamma_a$ is the specific gravity of new bitumen; and $\rho_w$ is the specific gravity of water.

3.1.2. Indirect Tensile Test

The indirect tensile test can be used to evaluate the cracking resistance of the bitumen mixture, which is normally used in studying the mechanical properties of CRMB. In China, the mechanical properties of the bitumen mixture at 15 °C can be used as parameters in pavement design [28]. Thus, the CRMB specimens were stored at 15 °C for 6 h in a temperature-controlled chamber after curing, and then, the indirect tensile test was performed at 15 °C. The loading displacement was applied at 50 mm/min through a (12.7 ± 0.1) mm-wide loading strip. The load and displacement of specimens were recorded during the whole loading process. Based on the maximum vertical force ($F$) and the corresponding horizontal displacement ($u$), the indirect tensile strength and failure strain can be calculated by Equations (3) and (4), which can evaluate the strength and ductility of CRMB.

$$ITS = \frac{2F}{\pi dh} \tag{3}$$

$$\varepsilon_{IT} = \frac{2u(1 + 3\mu)}{d(4 + \pi\mu - \pi)} \tag{4}$$

where $d$ is the specimen diameter; $h$ is the specimen height; and $\mu$ is Poisson's ratio assumed to be 0.3 according to the Chinese standard [3].

3.1.3. Moisture Susceptibility Test

Moisture susceptibility is one of the main properties for the durability of mixture. According to the Chinese specification [3], the immersion ITS and freeze–thaw ITS tests were conducted to evaluate the moisture damage resistance of CRMB.

Eight specimens, which were divided into two groups, were used in the immersion ITS test. The specimens in the first group were stored in a chamber with a constant temperature at 15 °C for 6 h before the ITS test. The other four specimens were immersed in a 25 °C water bath for 23 h and then immersed in a 15 °C water bath for 1 h before ITS test. The ITS test was performed at 15 °C with a constant loading rate of 50 mm/min. The wet to dry ITS ratio (TSR$_{wet/dry}$) was as an indication of the resistance of moisture damage.

Similarly to the immersion ITS test, eight specimens were also divided into two groups for the freeze–thaw ITS test. The first group was called the unconditioned group. Four specimens in this group were immersed in a water bath at 25 °C for 2 h before the test. The second group was called the conditioned group, in which specimens were treated as the following processes. Firstly, specimens were treated by vacuum saturation with vacuum degree between 97.3 and 98.7 kPa for 15 min; then, they were wrapped in plastic bags with 10 mL water. Subsequently, the specimens were exposed to a freeze–thaw cycle with a freezing condition at −18 °C for 16 h and a thawing condition at 60 °C for 24 h. Finally, the condition specimens were soaked for 2 h at 25 °C before the test. The ITS test of these eight specimens was performed at 25 °C with a constant loading rate of 50 mm/min. The ITS ratio of conditioned and unconditioned specimens (TSR$_{freeze-thaw}$) was calculated as an indication of moisture resistance. This indication is more rigorous than TSR$_{wet/dry}$ in evaluating the moisture resistance of the bitumen mixture [27].

### 3.1.4. Immersion Cantabro Test

CRMB has very high void content, and the adhesive ability of bitumen emulsion is relatively weak compared to hot bitumen. When its voids are filled with water, the phenomenon of moisture damage (i.e., stripping and shelling) is easily occurred under the tires load. However, the TSR$_{wet/dry}$ of CRMB is normally very high because the addition of cement is beneficial to this parameter. Therefore, the immersion Cantabro test was performed to further check the resistance of moisture damage [27].

The immersion Cantabro test is performed as the following processes. The specimens were firstly immersed in a 60 °C water bath for 48 h; then, they were taken out from the water bath and conditioned to 20 °C for 24 h. The specimens were firstly weighed after these treatments, and then, they were individually subjected to 300 revolutions (30–33 rpm) in a Los Angeles abrasion drum without steel charge. After the abrasion treatment, the specimens were weighed again. The mass loss ratio of specimens was calculated by Equation (5) as an indication of the moisture resistance of the mixture under the effect of abrasion. This indication is highly related to the adhesive ability of the bitumen binder.

$$\text{R}_{\text{ML}} = \frac{m_0 - m_1}{m_0} \times 100\% \tag{5}$$

where $\text{R}_{\text{ML}}$ is the mass loss ratio; $m_0$ and $m_1$ are the mass of specimen before and after abrasion.

### 3.1.5. Wheel Tracking Test

Rectangular slab specimens with dimensions of 30 cm × 30 cm × 5 cm were fabricated and cured at 60 °C for 2 days. After curing, the slab specimens were placed in the wheel tracking device at a constant temperature of 60 ± 0.5 °C for 6 hours. Then, the slab specimens were individually loaded for 60 min by a rubber tire with a contact pressure of 0.7 MPa and a tire traveling distance of 230 ± 10 mm. The tire-loading frequency was 42 ± 1 cycle/min. The rutting depth of specimens was measured by linear variable differential transformer (LVDT) to calculate the dynamic stability by Equation (6) for the rutting resistance evaluation.

$$DS = \frac{15N}{d_{60} - d_{45}} \tag{6}$$

where $DS$ is the dynamic stability; $N$ is the wheel traveling ($N$ = 42 cycle/min); $d_{60}$ and $d_{45}$ are the rutting depth at 60 and 45 min, respectively.

### 3.2. Mechanism Analysis Tests

3.2.1. Rejuvenating Evaluation Test of Aged Bitumen

An original basic bitumen with 80/100 penetration grade was gradually treated by a rolling-thin film oven (RTFO) and pressure-aging vessel (PAV). Then, the aged bitumen was blended with the rejuvenating agent. The rejuvenating agent to aged bitumen ratio was 1:4.8, which was the same as CRMB. After these treatments, the softening point, ductility, and penetration of basic bitumen, aged bitumen, and rejuvenated bitumen were tested to evaluate the rejuvenating results of the aged bitumen with the employed rejuvenating agent.

3.2.2. Microstructure Analysis

The microstructure of CRMB was investigated by scanning electron microscopy (SEM), including the microstructure of RAP, new cement bitumen composite binder, and their interface. Before the test, the surface of the tested specimens should be sprayed by a thin gold film to ensure a good conductivity for the SEM test.

3.2.3. Contact Angle Test

The wetting ability of bitumen emulsion and water on the surface of simulated RAP with aged and rejuvenated bitumen were investigated, respectively. Two types of simulated RAP with polished surface were fabricated. To fabricate the simulated RAP with a smooth surface, basalt aggregates were firstly cut and polished to obtain a smooth surface. Then, the polished aggregates were covered by aged and rejuvenated bitumen, respectively. The contact angle of bitumen emulsion and water on the surface of the two types of simulated RAP were tested by using an optical contact angle meter (SL200KB from Kino group, New York, NY, USA). The test was done by the sessile drop method at ambient temperature. The test was performed thrice for each specimen.

### 4. Results and Discussion

#### 4.1. Results of Pavement Properties of CRMB

4.1.1. Void Content of CRMB with Rejuvenating Agent

Figure 3 shows the effect of rejuvenating time on the void content of CRME with rejuvenating agent. It can be seen from Figure 3 that the void content of CRME with rejuvenating agent increases with the increasing rejuvenating time. Compared to reference CRME, CRME with a rejuvenating agent at 2 days has smaller void content. Therefore, RAP treated with a rejuvenating agent is more easily compacted with less rejuvenating time. This phenomenon can be explained as follows. The reactivity of rejuvenating agent with the aged bitumen is from the external to internal aged bitumen on the surface of RAP. When this reactivity is small, as an emulsion-based liquid, the rejuvenating agent can be a lubricant for RAP during compaction. However, when this reactivity is sufficient, there are no liquid rejuvenating agent on the external surface of RAP. As a result, the void content of CRMB can be increased. It is reasonable that the compactability of RAP treated with the rejuvenating agent decreases with the rejuvenating time.

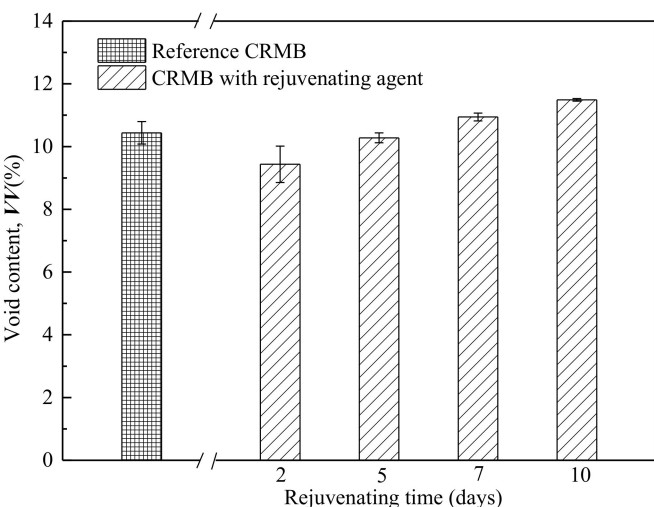

**Figure 3.** Void content of CRMB with rejuvenating agent at different rejuvenating times.

### 4.1.2. Indirect Tensile Strength of CRMB with Rejuvenating Agent

The ITS of CRMB with rejuvenating agent at different rejuvenating times is shown in Figure 4. A pleasant surprise can be seen from Figure 4 that the ITS of all CRMBs with rejuvenating agent are higher than that of reference CRMB. This phenomenon indicates that the addition of rejuvenating agent can be beneficial to the strength of CRMB, although the rejuvenating agent can soften the aged bitumen of RAP. Meanwhile, the adverse effect of the rejuvenating agent on the new bitumen in emulsion may be minimized by the treatment method of RAP in this study. The reasons for this surprise will be discussed in Section 4.2.

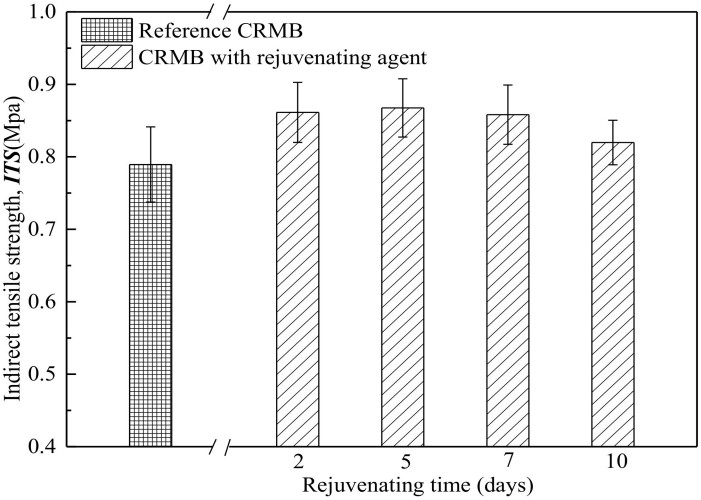

**Figure 4.** ITS of CRMB with rejuvenating agent at different rejuvenating times.

It can be also noted from Figure 4 that the ITS of CRMB with rejuvenating agent changes little when the rejuvenating time is from 2 to 7 days, but it begins to decrease with the rejuvenating time from 7 to 10 days. However, the void content of CRMB with rejuvenating agent in Figure 3 stably increases with the increasing rejuvenating time, which can be harmful to the ITS of CRMB. In comparison of the two opposite trends, it can be inferred that the aged bitumen of RAP treated by rejuvenating agent with more time may be more helpful to the ITS of CRMB if the void content does not change. However, in consideration of the adverse effect of the rejuvenating agent on the compactability of CRMB, the rejuvenating time should be no more than 7 days in order to obtain a good strength.

### 4.1.3. Failure Strain of CRMB with Rejuvenating Agent

The failure strain of CRMB with rejuvenating agent at different rejuvenating times is shown in Figure 5. As expected, the addition of rejuvenating agent can greatly increase the failure strain of CRMB, which is beneficial to the ductility and cracking resistance of CRMB. In addition, the failure strain of CRMB with rejuvenating agent increases with the increasing rejuvenating time, especially from 2 to 5 days. The rejuvenating agent can soften the aged bitumen and increase its ductility. However, the reactivity between the rejuvenating agent and aged bitumen is very gentle, thus requiring much time. In consideration of the failure strain, the rejuvenating time should be no less than 5 days. The cracking resistance of CRMB is related to both failure strain and ITS. In consideration of the two indications, the optimum rejuvenating time should be between 5 and 7 days.

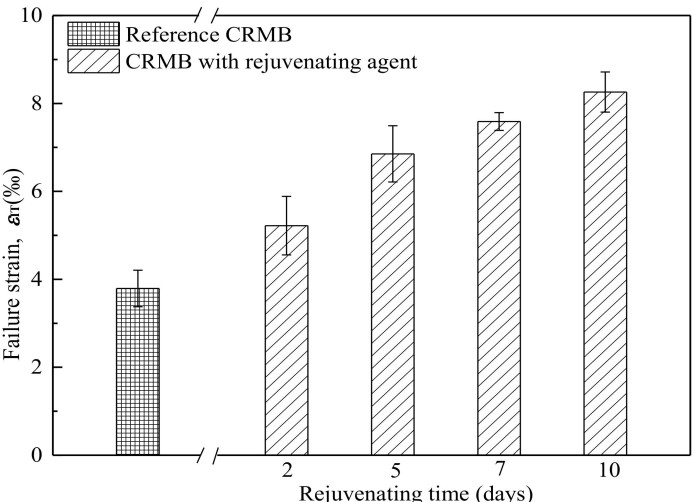

**Figure 5.** Failure strain of CRMB with rejuvenating agent at different rejuvenating times.

### 4.1.4. Moisture Susceptibility of CRMB with Rejuvenating Agent

The wet and dry ITS of CRMBs are shown in Figure 6. It can be seen from Figure 6 that although CRMBs with rejuvenating agent have slightly lower $TSR_{wet/dry}$ than the reference CRMB, all CRMBs with rejuvenating agent have a high value of $TSR_{wet/dry}$ (higher than 85%), indicating that all CRMBs can have good moisture susceptibility in this test condition.

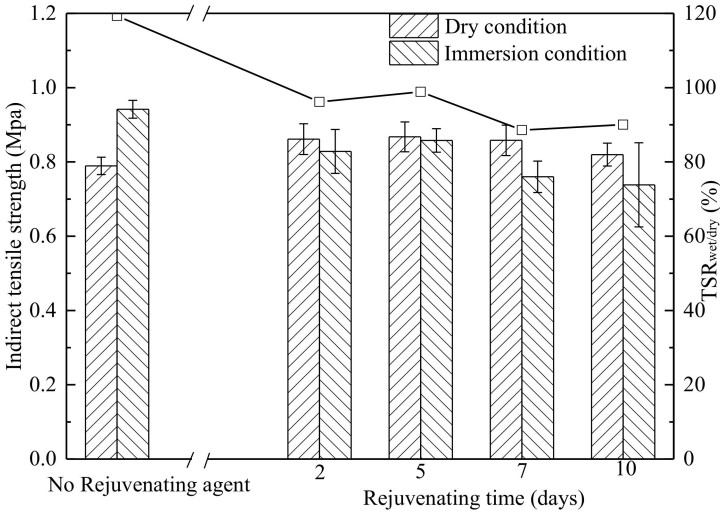

**Figure 6.** Wet and dry ITS of CRMB with rejuvenating agent at different rejuvenating times.

The unconditioned and conditioned ITS of CRMBs are shown in Figure 7. It can be seen from Figure 7 that the $\text{TSR}_{\text{freeze–thaw}}$ values of all CRMBs are much lower than the $\text{TSR}_{\text{wet/dry}}$ values. This is because the $\text{TSR}_{\text{freeze-thaw}}$ is more rigorous than the $\text{TSR}_{\text{wet/dry}}$ as an indication in evaluating the moisture susceptibility of the bitumen mixture. Compared to the reference CRMB, CRMB with rejuvenating agent can have a higher $\text{TSR}_{\text{freeze–thaw}}$ value. Thus, the addition of rejuvenating agent can improve the moisture susceptibility of CRMB. The unconditioned and conditioned ITS of CRMB with rejuvenating agent gradually decrease with the increasing rejuvenating time from 5 to 10 days. Except for CRMB with rejuvenating agent at 10 days, the unconditioned and conditioned ITS of all CRMBs with rejuvenating agent are higher than those of the reference CRMB. Therefore, the rejuvenating time should be no more than 7 days for a better moisture resistance and strength.

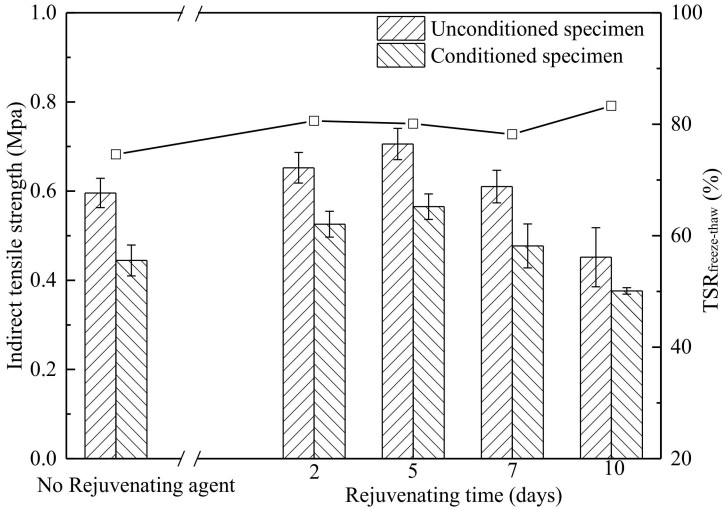

**Figure 7.** Unconditioned and conditioned ITS of CRMB with rejuvenating agent at different rejuvenating times.

The mass loss ratio of CRMB in the immersion Cantabro test is shown in Figure 8. It can be seen from Figure 8 that the addition of rejuvenating agent can reduce the mass loss ratio of CRMB. In addition, the mass loss rate of CRMB with rejuvenating agent decreases quickly with the increasing rejuvenating time from 2 to 5 days, and then, it changes little from 5 to 10 days. As mentioned previously, the mass loss ratio is an indication of the moisture susceptibility of the mixture under the effect of abrasion. It is also a more rigorous indication of moisture susceptibility than the $\text{TSR}_{\text{wet/dry}}$. Therefore, the rejuvenating time should be no less than 5 days from the indication of the mass loss ratio.

Overall, the addition of rejuvenating agent can improve the moisture susceptibility of CRMB. In comparison of the above three indications, the optimum range of rejuvenating time for the rejuvenating agent should be between 5 and 7 days. Since all CRMBs have the same aggregate gradation and their void content differs slightly, the difference of the moisture susceptibility is mainly due to the cohesive and adhesive ability of binder, including the aged bitumen, fresh bitumen mastic, and their interface, which will be discussed in the following Section 4.2.

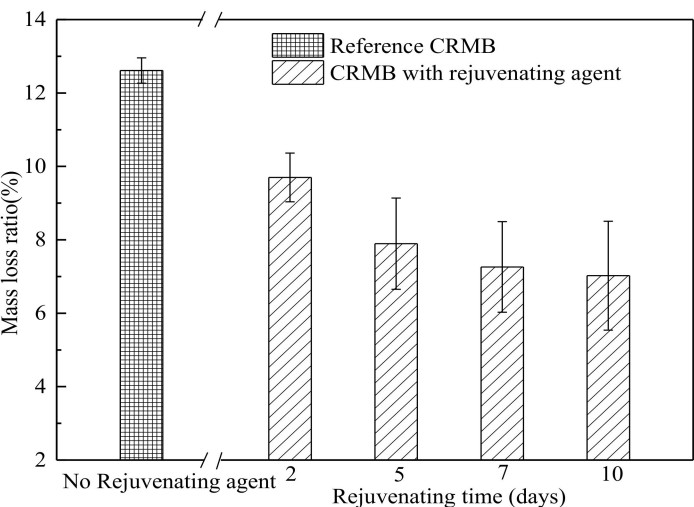

**Figure 8.** The mass loss ratio of CRMB in an immersion Cantabro test.

### 4.1.5. Rutting Resistance of CRMB with Rejuvenating Agent

The dynamic stability of CRMB with and without rejuvenating agent is shown in Figure 9. It can be seen from Figure 9 that the dynamic stability of CRMB is slightly decreased with the addition of rejuvenating agent. Therefore, the rejuvenating agent is slightly harmful to the rutting resistance of CRMB because it can soften the aged bitumen. However, the dynamic stability of CRMB is very high, because the addition of cement can greatly reduce the temperature susceptibility of bitumen binder [29]. Thus, CRMB with rejuvenating agent still has good rutting resistance.

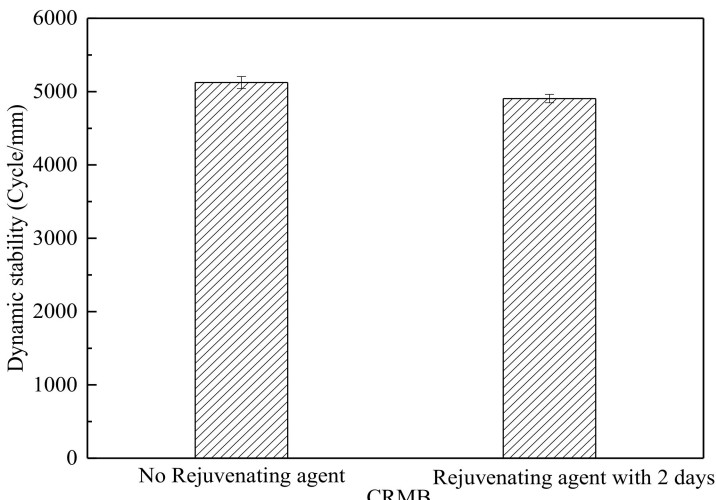

**Figure 9.** Dynamic stability of CRMB.

### 4.2. *Analysis of The Improvement Mechanisms of Pavement Properties*

#### 4.2.1. Effect of Rejuvenating Agent on The Basic Properties of Aged Bitumen

The softening point, ductility, and penetration of basic bitumen, aged bitumen, and rejuvenated bitumen were tested to evaluate the rejuvenating results of the aged bitumen with the employed rejuvenating agent, and the results are shown in Table 2. The rejuvenating time is 2 days. It can be seen from Table 2 that the rejuvenating agent can restore the technical properties of the aged bitumen, e.g., increasing the penetration and ductility and decreasing the softening point. As a result, the addition of rejuvenating agent can increase the ductility of CRMB but decrease the rutting resistance of CRMB. It should be noted that

the basic technical properties of rejuvenated bitumen are almost equal to those of original basic bitumen. Thus, the rejuvenating agent content is very suitable.

**Table 2.** Basic properties of bitumen before and after rejuvenation.

| Bitumen Type | Penetration (25 °C, 0.1 mm) | Softening Point (°C) | Ductility (15 °C, cm) |
|---|---|---|---|
| Original basic bitumen | 95.3 | 47.7 | ≥100 |
| Aged bitumen | 60 | 50.6 | 67.9 |
| Rejuvenated bitumen | 87.6 | 48.1 | ≥100 |

### 4.2.2. Effect of Rejuvenating Agent on The Microstructure of CRMB

The microstructures of the bitumen emulsion mastic and the interface between mastic and RAP were investigated to reveal the effect of rejuvenating agent on the pavement properties of CRMB. The microstructure of bitumen emulsion mastic with and without rejuvenating agent is shown in Figure 10. It can be seen from Figure 10a that many pores can be observed in the bitumen emulsion mastic of reference CRMB, which affects the continuity of the bitumen membrane. However, for CRMB treated with rejuvenating agent for 2 days in Figure 10b, its bitumen emulsion mastic has a continual membrane structure with few pores. Therefore, the addition of rejuvenating agent can be beneficial to forming a denser microstructure for bitumen emulsion mastic. Previous studies indicate that the pores in bitumen emulsion mastic are formed due to water evaporation [9,23]. The water content of CRMB with rejuvenating agent is less than that of the reference CRMB. As a result, the addition of rejuvenating agent can reduce the pores number of bitumen emulsion mastic. The rejuvenating agent can also soften the new bitumen in bitumen emulsion mastic, which is also beneficial to the formation of bitumen membrane. As a result of these two effects, the microstructure of mastic in Figure 10b is much denser than that in Figure 10a.

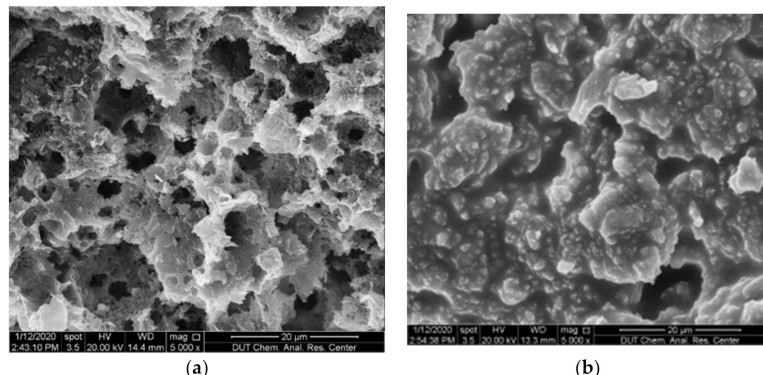

(**a**)  (**b**)

**Figure 10.** The microstructure of mastic in CRMB: (**a**) untreated; (**b**) treated with 2 days.

The interface microstructure between mastic and RAP in CRMB with and without rejuvenating agent is shown in Figure 11. There are two different morphologic parts for each image. A continual and smooth surface is observed in a part for each image, which indicates that the part is RAP with a bitumen surface. The other part is a relative rough surface with a thinner bitumen membrane. It can be inferred that this part is cement bitumen emulsion mastic. It can be seen from Figure 11a that a microcrack can be observed in the interface between bitumen emulsion mastic and RAP in the reference CRMB. Thus, the bitumen emulsion mastic cannot physically bond well with RAP in CRMB. However, CRMB with a rejuvenating agent has a well-physical bonding interface between bitumen emulsion mastic and RAP in Figure 11b. Therefore, the rejuvenating agent can greatly improve the interface between bitumen emulsion mastic and RAP. This improvement in the interface is probably due to the rejuvenating agent on the external surface of aged bitumen being able to react with the new bitumen in the bitumen emulsion mastic.

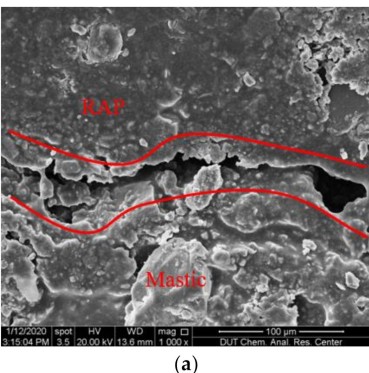
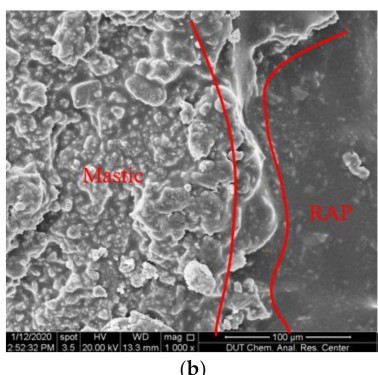

(**a**)                (**b**)

**Figure 11.** The interface microstructure between mastic and RAP in CRMB: (**a**) untreated; (**b**) treated with 2 days.

It can be also noted in Figure 11 that the addition of rejuvenating agent can improve the membrane structure of bitumen on RAP. The surface of the RAP in CRMB with rejuvenating agent is smoother than that in the reference CRMB because most of the rejuvenating agent reacts with aged bitumen and improves the coating of aged bitumen on old aggregate.

Overall, the rejuvenating agent not only restores the technical properties of aged bitumen but also improves the membrane structure of bitumen emulsion mastic and aged bitumen. It can also greatly improve the bonding interface between RAP and bitumen emulsion mastic. As a result, although the rejuvenating agent can soften bitumen, its addition can still slightly increase the ITS of CRMB, but it has little effect on the rutting resistance of CRMB. Since it can greatly improve the membrane structure of bitumen emulsion mastic and aged bitumen, its addition can greatly improve the ductility of CRMB.

4.2.3. Effect of Rejuvenating Agent on The Wetting Ability of Bitumen Emulsion on RAP

It was discussed previously in Figure 11 that the rejuvenating agent can greatly improve the bonding interface between bitumen emulsion mastic and RAP. There are two possible reasons that account for this phenomenon. The first reason is the rejuvenating agent on the external surface of aged bitumen can react with the new bitumen in the bitumen emulsion mastic, which was mentioned previously. The second reason may be that the wetting ability of bitumen emulsion on RAP in the fresh state may be improved by the rejuvenating agent. To confirm the second reason, the contact angle of bitumen emulsion and water on the simulated RAP with a polished surface was investigated, which is shown in Figure 12. It can be seen from Figure 12 that the contact angle of bitumen emulsion and water on the simulated RAP decreases with the rejuvenating time. The addition of rejuvenating agent can improve the wetting ability of bitumen emulsion on the surface of RAP, which is beneficial to the bonding interface between bitumen emulsion mastic and RAP. This phenomenon is probably related to the composition of the rejuvenating agent. To better ensure a sufficient reactivity between the rejuvenating agent and aged bitumen, a wetting agent is normally added into the rejuvenating agent to increase its penetrative ability into aged bitumen. This wetting agent can be also beneficial to the wetting ability of bitumen emulsion and water on the treated RAP.

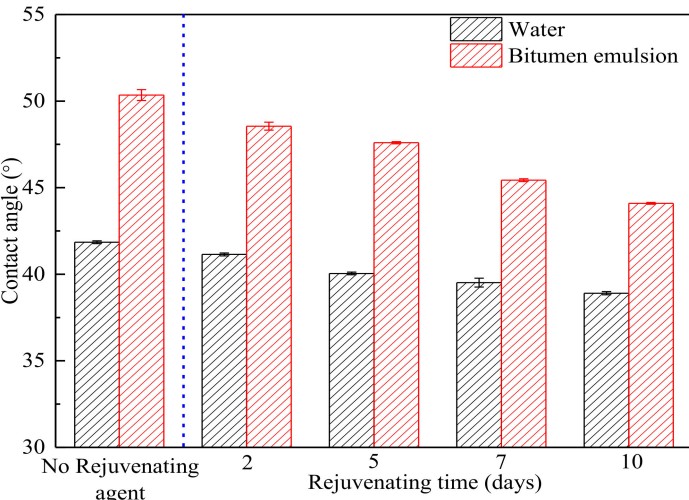

**Figure 12.** Contact angle of bitumen emulsion and water on the simulated RAP.

## 5. Conclusions

The effect of rejuvenating agent and its rejuvenating time with RAP on the pavement properties of CRMB were investigated. The improvement mechanisms of the rejuvenating agent on the pavement properties of CRMB were discussed. Based on the results and discussion, the following conclusions can be drawn:

1. The addition of rejuvenating agent can slightly increase the indirect tensile strength of CRMB and greatly increase the failure strain of CRMB. It can also increase the $TSR_{freeze-thaw}$ value of CRMB and decreases the mass loss ratio of CRMB in an immersion Cantabro test. Therefore, the addition of rejuvenating agent can be beneficial to the cracking resistance and moisture susceptibility of CRMB.
2. Since the reactivity between the rejuvenating agent and RAP at ambient temperature requires a certain time; thus, the rejuvenating time can affect the pavement properties of CRMB. Specifically, the indirect tensile strength of CRMB changes little with the rejuvenating time from 2 to 7 days, but it then decreases from 7 to 10 days. The failure strain of CRMB increases significantly from 2 to 5 days, and then, it increases slightly from 5 to 10 days. The mass loss ratio of CRMB in an immersion Cantabro test decreases quickly from 2 to 5 d, and then, it decreases slightly from 5 to 10 days. After comparison of the above pavement properties, the optimum rejuvenating time should be between 5 and 7 days.
3. Although the rejuvenating agent can soften aged bitumen, the addition of rejuvenating agent can still increase the indirect tensile strength of CRMB. Meanwhile, its addition does not greatly reduce the rutting resistance of CRMB. The phenomenon can be attributed to that the rejuvenating agent can be both beneficial to the membrane structure of bitumen emulsion mastic and aged bitumen. It can also greatly improve the bonding interface between RAP and bitumen emulsion mastic because it decreases the contact angle of bitumen emulsion on the simulated RAP. Therefore, the addition of rejuvenating agent can be a good choice to improve the pavement properties of CRMB.

**Author Contributions:** Conceptualization, J.O.; methodology, C.L.; formal analysis, C.L.; investigation, C.L. and J.S.; data curation, P.C., W.Y. and Y.S.; writing—original draft preparation, C.L., W.Y. and Y.S.; writing—review and editing, J.O. and P.C.; supervision, J.O. and P.C.; funding acquisition, J.O. All authors have read and agreed to the published version of the manuscript.

**Funding:** The authors thank the Natural Science Foundation of Liaoning Province (2020-MS-116) and the National Natural Science Foundation of China (51608096).

**Institutional Review Board Statement:** Not applicable.

**Informed Consent Statement:** Not applicable.

**Data Availability Statement:** Data is contained within the article.

**Conflicts of Interest:** The authors declare that they have no conflict of interest.

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
