# Peer review of "Effect of Rejuvenating Agent on the Pavement Properties of Cold Recycled Mixture with Bitmen Emulsion"

_coatings, doi:10.3390/coatings11050520_

Round 1
Reviewer 1 Report
The manuscript presents the effect of rejuvenating agent on the pavement properties of CRMB along with its treatment process. The paper is interesting. Below are the recommendations:
- As rejuvenating agent is the main component of the paper that should be describe in more details. Its physical, chemical properties and so on needs to be described in details. I would highly suggest to provide a table which also showing the difference of that rejuvenating agent compared to the rejuvenating agent that used in hot recycling asphalt. Also, provide a image of the rejuvenating agent as well.
- How (or what basis) the reactivity time was chosen?
- One thing missing in the results and discussions section is that the quantitative comparisons (discussions are made qualitatively). The term such as "greatly/slightly" does not tell readers how effective it is. I would strongly suggest authors to add quantitative comparisons in the results and discussions section.
- Figure 12. Please give different pattern type to any one of the bar.
- Right now. the abstract and conclusion part are generic. With inclusions of quantitative discussions in the "results and discussions' part, abstract and conclusions should be revised.
Reviewer 2 Report
The following comments are expected to improve the manuscript:
English language and format suggestions:
- Effect of rejuvenating agent on the pavement properties of cold recycled mixture with bitmen
- In the abstract: To obtain better pavement properties, the optimum rejuvenating time is recommended between 5 d and 7 d (days should be used).
- Table 1: One column should be added to display the unit.
- Figure 1: the upper and lower limit line should be modified into different line styles.
Citations should be added in the following statements:
- The cement content of CRMB should be between 1% and 2% to prevent cracking in the application.
Other comments:
- The abstract is suggested to be concise and focus on the research objective.
- The author state that: “thus the RAP is normally as black aggregates in the traditional CRMB.” Further description of the black aggregate can be provided.
- The addition of some research about Cement Asphalt Mortar and the usage of Asphalt Emulsion is expected to clarify the role of emulsion in the literature review.
- The properties of Rejuvenator are suggested to compare with conventional rejuvenator of hot recycling asphalt, especially the viscosity. Table is a recommended tool.
- “Thus, the RAP was firstly blended well with the rejuvenating agent and then kept at 25oC for several days.” How to blend the RAP with the rejuvenating agent?
- Besides, it can be seen from Fig.2 that the addition of rejuvenating agent can improve the compactability of CRMB. How to evaluate the compactability through Fig.2 without referring to the explanation in Fig.3?
- The rutting resistances of remained curing day conditions should be provided in Figure 9.
- Figure 10 and the associated description provide very good insight into the effect of rejuvenating agent on the pavement properties of CRMB. Therefore, the arrows and legends system can be used to improve the Figure 10 content.
Comment: Minor Revision before Published.

Reviewer 3 Report
- It is hard to review a paper without line numbers.
- Please improve the abstract. It is very hard to read. Make it simple.
- You need to improve the language. Probably some native speaker should read it.
- You repeating some parts of the manuscript like "rejuvenating agent should be firstly blended with RAP" or "ambient temperature is much gentler than those at high temperature". It is annoying when you have to read it for the nth time.
- You are not testing pavement in this research.
- How have you blended rejuvenator with rap?
- You should at least mention how are you going to blend rejuvenator with rap in the field condition.
Reviewer 4 Report
Dear authors,
I appreciated your research regarding the effects of rejuvenating agents on CRMB. The article is well structured and supported by an extensive experimental campaign. I also appreciated the effort to give a solid explanation at your results. Please find below some little comments and revisions that I hope you will take into consideration:
- Page 2, line 5 the term “surface layer” could be confusing, because after you talk about intermediate and bottom layer. Please revise this sentence;
- Page 3, a scheme could help the reader to better understanding the preparation of your CRMB;
- In figure 1, do upper and lower limits refer to any standard? If yes, please indicate it;
- What is the role of water in CRMB? Reducing the friction between the aggregates in RAP during compaction? Please explain it?
- The line with little squares of Figure 6 indicates the ratio TSRwet/dry? If yes, please add it in the legend;
- Same for figure 7;
- In chapter 4.1.5. did you perform the test also at 5, 7 and 10 days? If yes, please add the results in figure 9;
- In conclusions, perspectives and/or future researches are missing. Can you add them?
Round 2
Reviewer 1 Report
The authors addressed the reviewer comments.
Reviewer 2 Report
The revised paper satisfies my inquiries and the response from the reviewer clarifies the unclear problems. Therefore, I approve this paper for publication.
Reviewer 4 Report
Dear authors,
thanks to take into account my suggestions. I think that the figure 3 is very helpful for better understanding the preparation process.